# Suppression of retinal degeneration by two novel ERAD ubiquitin E3 ligases SORDD1/2 in *Drosophila*

Jaiwei Xu[1,2], Haifang Zhao[2], Tao Wang[2,3]*

1 College of Biological Sciences, China Agricultural University, China, 2 National Institute of Biological Sciences, China, 3 Tsinghua Institute of Multidisciplinary Biomedical Research, Tsinghua University, China

* wangtao1006@nibs.ac.cn

## Abstract

Mutations in the gene rhodopsin are one of the major causes of autosomal dominant retinitis pigmentosa (adRP). Mutant forms of Rhodopsin frequently accumulate in the endoplasmic reticulum (ER), cause ER stress, and trigger photoreceptor cell degeneration. Here, we performed a genome-wide screen to identify suppressors of retinal degeneration in a *Drosophila* model of adRP, carrying a point mutation in the major rhodopsin, Rh1 (Rh1$^{G69D}$). We identified two novel E3 ubiquitin ligases SORDD1 and SORDD2 that effectively suppressed Rh1$^{G69D}$-induced photoreceptor dysfunction and retinal degeneration. SORDD1/2 promoted the ubiquitination and degradation of Rh1$^{G69D}$ through VCP (valosin containing protein) and independent of processes reliant on the HRD1 (HMG-CoA reductase degradation protein 1)/HRD3 complex. We further demonstrate that SORDD1/2 and HRD1 function in parallel and in a redundant fashion to maintain rhodopsin homeostasis and integrity of photoreceptor cells. These findings identify a new ER-associated protein degradation (ERAD) pathway and suggest that facilitating SORDD1/2 function may be a therapeutic strategy to treat adRP.

## Author summary

Misfolded rhodopsins accumulated in endoplasmic reticulum (ER) could disrupt the homeostasis of the ER and cause ER stress. Chronic ER stress would finally lead to photoreceptor cell death and retinal degeneration. To diminish the stress and sustain homeostasis cells develop alternative strategies to clear the misfolded rhodopsins. Previous studies have suggested that ubiquitin E3 ligase HRD1 is involved in the degradation of misfolded rhodopsins. In this study, we define novel ubiquitin E3 ligase SORDD1/2 based on a genetic screen and demonstrate that SORDD1/2 promotes the degradation of misfolded rhodopsins through ER-associated degradation (ERAD) pathway. Furthermore, we demonstrate that SORDD1/2 function independently of HRD1 in misfolded rhodopsins degradation. We also show SORDD1/2 and HRD1 play redundant roles in rhodopsin homeostasis. Finally, we demonstrate that SORDD1 works well in a *Drosophila* disease model. Our studies identify a novel ERAD pathway that acts in parallel to HRD1, and suggest that SORDD1 is a good candidate therapeutic target.

**Data Availability Statement:** All relevant data are within the manuscript and its Supporting Information files.

**Funding:** This work was supported by grants from the National Natural Science Foundation of China

(81670891 and 81870693) awarded to T. Wang. The funders had no role in study design, data collection and analysis, decision to publish, or preparation of the manuscript.

**Competing interests:** The authors have declared that no competing interests exist

## Introduction

Misfolded membrane proteins, including G protein-coupled receptors (GPCRs), are often retained in the endoplasmic reticulum (ER) and an overabundance of ER-retained proteins is associated with many diseases. Rhodopsin is a GPCR and dominant mutations in the rhodopsin gene (Rho) are observed in 20–30% of all forms of autosomal dominant retinitis pigmentosa (adRP), the most common form of retinal degeneration [1–3]. Most biochemically-characterized Rho mutants associated with adRP are likely misfolded and accumulate in the ER of photoreceptor neurons [4]. Indeed, the Rho mutation most commonly associated with adRP is a proline to histidine mutation at position 23 (Rho[P23H]), which leads to Rho retention within the ER. This in turn leads to ER stress, activation of the unfolded protein response (UPR), and ultimately photoreceptor degeneration [2, 5–8]. This is seen in both human patients and animal models of the disease.

In most cases, misfolded rhodopsins are inherently unstable and undergo ER-associated protein degradation (ERAD), a process by which substrates are recognized, ubiquitinated by E3 ligases, retro-translocated into the cytosol, and sent to proteasomes for degradation [8–11]. The UPR induces ERAD to promote the degradation of misfolded rhodopsin and to protect photoreceptor cells against ER stress signals [12–14]. Although it has been proposed that ERAD may disrupt Rho homeostasis earlier in development and contribute to retinal degeneration [15], studies in both *Drosophila* and mouse models suggest that increasing degradation of misfolded rhodopsin through ERAD and the proteasome is protective. Further, decreased efficiency of ERAD/proteasome function leads to the aggregation of rhodopsin and photoreceptor death [16–19]

The E3 ubiquitin-protein ligase HRD1 (HMG-CoA reductase degradation protein 1) plays a central role in ERAD of membrane proteins (ERAD-M) such as rhodopsin. In a fly model of adRP, a glycine is mutated to an aspartate at position 69 within Rh1, the major rhodopsin in flies, which is encoded by the *ninaE* locus (*ninaE[G69D]*) [20, 21]. Importantly, overexpression of HRD1 reduces Rh1[G69D]-induced ER stress and alleviates photoreceptor cell loss [19]. Despite the key role played by HRD1 in maintaining rhodopsin homeostasis, flies lacking HRD1 have normal rhodopsin and photoreceptor cell function [16], suggesting that another E3 ubiquitin ligase may help maintain rhodopsin homeostasis.

Here we screened for additional genes that reduce Rh1[G69D]-induced photoreceptor cell degeneration, and identified two E3 ubiquitin ligases, SORRD1 and SORRD2, that strongly suppress retinal degeneration in Rh1[G69D]-expressing flies. We demonstrate that SORRD1/2 ubiquitinates misfolded Rh1 and promotes degradation of these proteins through valosin containing protein (VCP) and the proteasome, and that this process is completely independent of HRD1. Moreover, although null mutations in either *sordd1/2* or *hrd1* do not lead to photoreceptor cell dysfunction, animals in which both *sordd1/2* and *hrd1* are mutated exhibit severe retinal degeneration. Overall, this work identifies an important role for the previously uncharacterized E3 ubiquitin ligases, SORRD1 and SORRD2, in ERAD of misfolded rhodopsin and maintenance of rhodopsin homeostasis, independent of HRD1.

## Results

### *SORDD1* and *SORDD2* are new suppressors of Rh1[G69D]

To find factors other than HRD1 that limit retinal degeneration in adRP, we performed a gain-of-function screen using a *Drosophila* model in which Rh1[G69D] was ectopically overexpressed using the *GMR-Gal4* system (*GMR>Rh1[G69D]*) (Fig 1A–1D and S1A–S1B Fig). Overexpression of HRD1, the major E3 ubiquitin ligase of the ERAD pathway, ameliorated the glossy eye phenotype of *GMR>Rh1[G69D]* flies and restored ommatidia structures (based on SEM analysis),

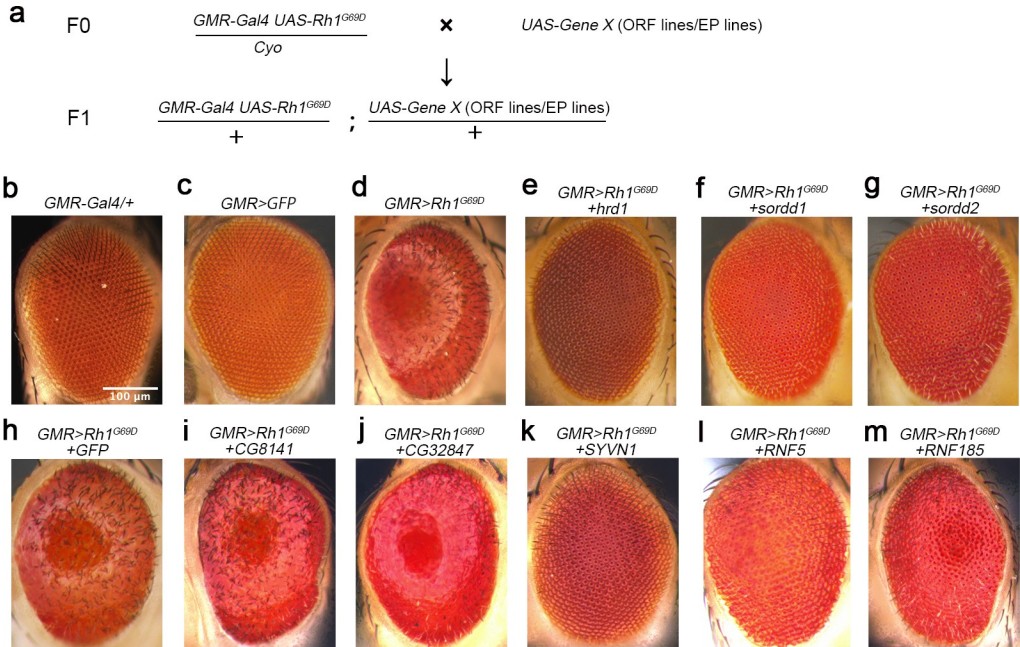

**Fig 1. _SORDD1_ and _SORDD2_ suppress Rh1$^{G69D}$-induced retinal degeneration.** (a) The genetic screen used to identify suppressors of Rh1$^{G69D}$-induced retinal degeneration. The representative strategy of screening for UAS-cDNAs inserted on the 3rd chromosome is shown. (b-m) Light photomicrographs show the adult eye morphology of (b) _GMR-gal4/+_, (c) _GMR-Gal4>GFP_ (_GMR-Gal4/+;UAS-GFP/+_) (d) _GMR>Rh1$^{G69D}$_ (_GMR-Gal4 UAS-Rh1$^{G69D}$/+_), (e) _GMR>Rh1$^{G69D}$+hrd1_ (_GMR-Gal4 UAS-Rh-1$^{G69D}$/+;UAS-hrd1/+_), (f) _GMR>Rh1$^{G69D}$+sordd1_ (_GMR-Gal4 UAS-Rh-1$^{G69D}$/+;UAS-sordd1/+_), (g) _GMR>Rh1$^{G69D}$+sordd2_ (_GMR-Gal4 UAS-Rh-1$^{G69D}$/+;UAS-sordd2/+_), (h) _GMR>Rh1$^{G69D}$+GFP_ (_GMR-Gal4 UAS-Rh-1$^{G69D}$/+;UAS-GFP/+_), (i) _GMR>Rh1$^{G69D}$+CG8141_ (_GMR-Gal4 UAS-Rh-1$^{G69D}$/+;UAS-CG8141/+_), (j) _GMR>Rh1$^{G69D}$+ CG32847_ (_GMR-Gal4 UAS-Rh-1$^{G69D}$/+;UAS-CG32847/+_), (k) _GMR>Rh1$^{G69D}$+SYVN1_ (_GMR-Gal4 UAS-Rh-1$^{G69D}$/+;UAS-SYVN1/+_), (l) _GMR>Rh1$^{G69D}$+RNF5_ (_GMR-Gal4 UAS-Rh-1$^{G69D}$/+;UAS-RNF5/+_) and (m) _GMR>Rh1$^{G69D}$+RNF185_ (_GMR-Gal4 UAS-Rh-1$^{G69D}$/+;GMR-RNF185/+_). Scale bar, 100 μm.

suggesting the viability of this genetic screen (Fig 1D and 1E, S1B, S1C and S1K Fig and S1 Data).

We screened ~3,000 UAS-cDNA libraries from FlyORF [22] and EP collections, and identified three genes that strongly suppressed Rh1$^{G69D}$-induced retinal degeneration. These were _smg5_ (_UAS-Smg5.ORF_), _CG8974_ (_P[EP]CG8974$^{G757}$_), and _CG32581_ (_P[EPgy2]EY20019_). Since Smg5 may affect the stability of _rh1$^{G69D}$_ mRNA, we focused on _CG8974_ and _CG32581_, which each encode RING-domain E3 ubiquitin ligases. We verified that _CG8974_ and _CG32581_ could each suppress Rh1$^{G69D}$-induced retinal degeneration by repeating the experiment using _UAS-CG8974_ or _UAS-CG32581_ overexpression lines (Fig 1H–1G, S1D, S1E and S1K Fig). We therefore named these proteins SORDD1 (Suppression Of Retinal Degeneration Disease 1 upon overexpression) and SORDD2, respectively. SORDD1 and SORDD2 share 92% protein sequence identity. RNA-sequencing and qPCR results indicated that while _sordd1_ is expressed ubiquitously, _sordd2_ is not expressed under normal conditions (S2B Fig, S2 Data). The _sordd1_ and _sordd2_ gene loci share 95% sequence identity and are adjacent in the genome (S2A Fig). This suggests that _sordd2_ may have resulted from gene duplication, and therefore may serve a largely, if not entirely, redundant role.

Phylogenetic analysis indicated that _sordd1 and sordd2_, as well as the genes _CG8141_ and _CG32847_, each encode E3 ligases that contain both a RING-finger motif and a transmembrane domain [23]. We overexpressed _CG8141_ or _CG32847_ in the _GMR>Rh1$^{G69D}$_ retina, but they failed to prevent Rh1$^{G69D}$-induced retinal degeneration, indicating a specific function for SORDD1/2 in degrading misfolded Rh1$^{G69D}$ (Fig 1I and 1J, S1F,S1G and S1K Fig, S1 Data).

Moreover, the human homologs of SORDD1/2 or HRD1 (namely RNF5, RNF185, or SYVN1, respectively) each suppressed retinal cell degeneration in $GMR>Rh1^{G69D}$ flies, indicating conservation of function for these E3 ligases (Fig 1K–1M and S1H–S1K Fig).

Previous studies reported that HRD1 can also recognize misfolded versions of wild-type Rh1, which are expressed in larval eye imaginal discs and earlier pupa eyes before the expression of chaperons for Rh1, and suppresses ER stress caused by this earlier expression of wild-type Rh1 [19]. We also found that SORDD1 and SORDD2 reduced retinal cell degeneration induced by the expression of wild-type Rh1 in larval eye imaginal discs (S2E Fig). Moreover, we expressed Rh1$^{P37H}$ (the equivalent of mammalian Rho$^{P23H}$ –the most common genetic mutation associated with ADRP) in larval eye imaginal discs using $GMR\text{-}Gal4/UAS\text{-}rh1^{P37H}$ and found that SORDD1 and SORDD2 largely suppressed Rh1$^{P37H}$-mediated retinal damage [24] (S2E Fig). These results suggest that SORDD1/2 and HRD1 recognize the folding state of Rh1 rather than specific alterations/mutations in the Rh1 protein sequence.

### The E3 ubiquitin ligases SORDD1/2 promote the degradation of Rh1$^{G69D}$

Overexpression SORDD1/2 associate with the membrane fraction of S2 cells and colocalize with the ER protein Calnexin (S2C and S2D Fig). Moreover, RNF185, the human homolog of SORDD1/2 was found to be an ER resident E3 ligase and function in the degradation of transmembrane proteins in ER, suggesting that SORDD1/2 may be directly involved in degrading misfolded Rh1$^{G69D}$ in ER [25, 26]. To test this, we expressed a Myc-tagged version of Rh1$^{G69D}$ in the eye ($GMR>Rh1^{G69D}\text{-}Myc$), and found that overexpression of SORDD1 or SORDD2 could lower the level of Rh1$^{G69D}$ protein in both eye imaginal discs and pupa eyes (Fig 2A–2D, S3 Data and S4 Data). Accumulation of Rh1$^{G69D}$ induced ER stress and activated the UPR, as both Xbp1-EGFP and ATF4-mCherry, two independent UPR reporters, were activated in eye imaginal discs of $GMR>Rh1^{G69D}$ animals [19, 27]. However, Rh1$^{G69D}$ failed to activate Xbp1-EGFP and ATF4-mCherry when SORDD1 or SORDD2, but not CG8141, was overexpressed (Fig 2A and 2B and S3 Data). Thus, like HRD1, SORDD1/2 facilitates the degradation of misfolded Rh1$^{G69D}$, suppressing Rh1$^{G69D}$-induced ER stress and retinal degeneration.

Phylogenetic analysis indicated that SORDD1 and SORDD2 belong to conserved RING-finger motifs and transmembrane regions E3 ligase family, we then introduced a single mutation into the conserved RING finger domain, changing Cys-165 to Ser to potentially block enzymatic activity (SORDD1$^{C165S}$ and SORDD2$^{C165S}$) (S2A Fig). As a positive control, we also mutated $hrd1$ ($hrd1^{C327S}$). As expected, SORDD1$^{C165S}$, SORDD2$^{C165S}$, and Hrd1$^{C327S}$ did not suppress Rh1$^{G69D}$-induced retinal degeneration (Fig 2E). We then mutated all the potential ubiquitylated sites (cytosol and luminal lysine residues) within Rh1$^{G69D}$ to generate a ubiquitylation-resistant form (Rh1$^{G69D}$ $^{+KR}$). Eye damage was more severe in Rh1$^{G69D+KR}$ flies compared to Rh1$^{G69D}$ (S3A Fig). Importantly, overexpression of SORDD1/2 or HRD1 did not fully rescue Rh1$^{G69D+KR}$-induced cell damage (S3A Fig). The partial rescue effect might be due to the three lysine residues remaining within the transmembrane domain of Rh1$^{G69D+KR}$ which could also be ubiquitylated by SORDD1/2. We next examined if Rh1$^{G69D}$ is the direct substrate of SORDD1 by immunoprecipitation of Rh1$^{G69D}$-Myc. We found that the expression of SORDD1 but not SORDD1$^{C165S}$ significantly increased levels of ubiquitinated Rh1$^{G69D}$-Myc (Fig 2F and 2G and S5 Data). Taking together, these data indicate that SORDD1/2 function as E3 ubiquitin ligases to ubiquitylate Rh1$^{G69D}$ *in vivo*.

### VCP and the proteasome but not the HRD1 complex are required for SORDD1-mediated degradation of Rh1$^{G69D}$

To identify factors that function upstream or downstream of SORDD1/2, we conducted a genome-wide RNAi screen for lines that abolished the suppressive effects of SORDD1 on

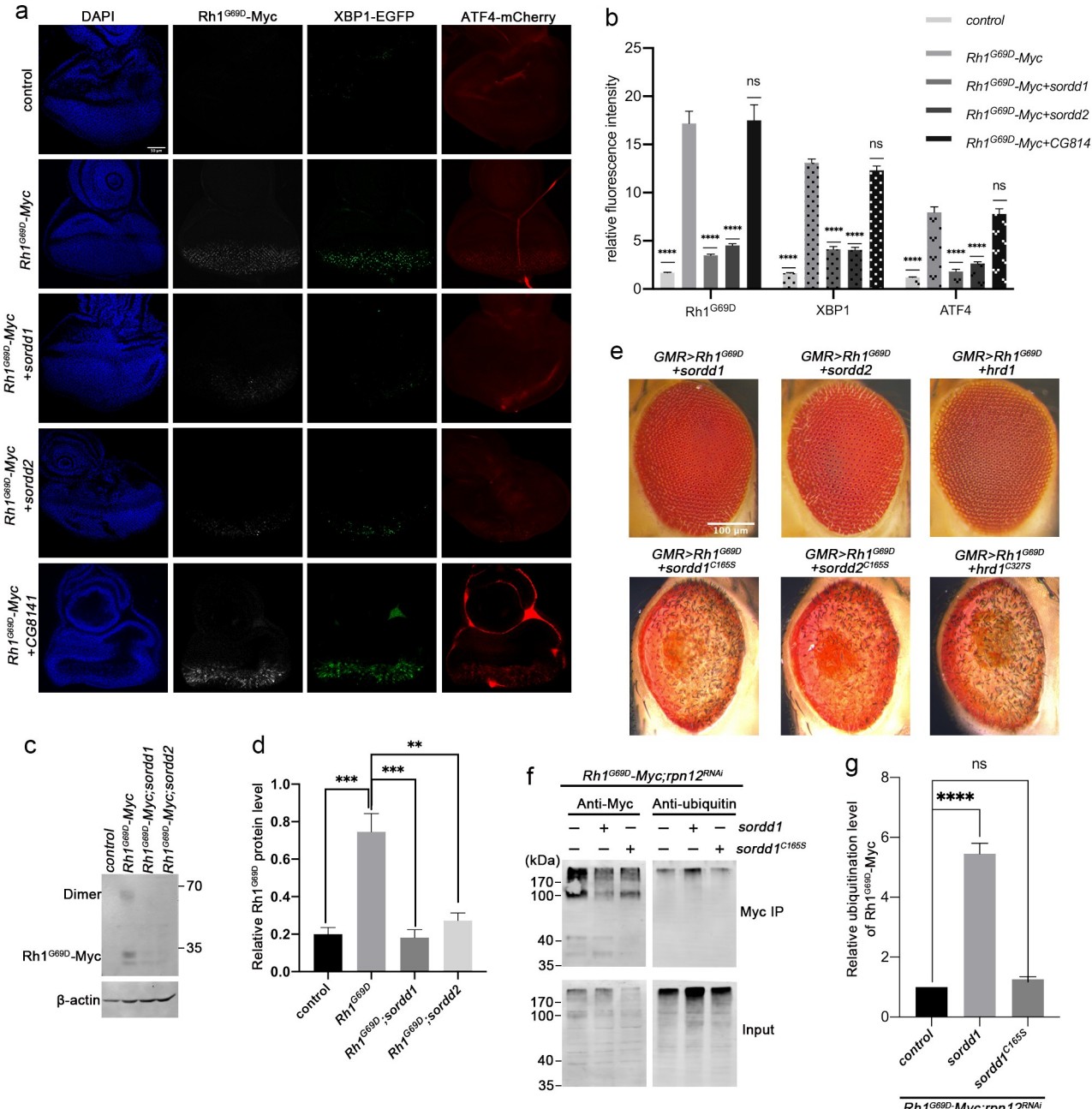

**Fig 2. The E3 ubiquitin ligases SORDD1/2 reduce the UPR by promoting the degradation of Rh1$^{G69D}$.** (a) Fluorescence images of eye imaginal discs. The ER stress markers, Xbp1-EGFP and ATF4-mCherry (control, *GMR-gla4 UAS-Xbp1 ATF4-mCherry/+*), were induced by the expression of Rh1$^{G69D}$–Myc (*Rh1$^{G69D}$–Myc, GMR-gal4 UAS Xbp1 ATF4-mCherry UAS-Rh1$^{G69D}$–Myc/+*). Expressing SORDD1 (*Rh1$^{G69D}$–Myc+sordd1, GMR-gla4 UAS-Xbp1 ATF4-mCherry UAS-Rh1$^{G69D}$–Myc/+;UAS-sordd1/+*) or SORDD2 (*Rh1$^{G69D}$–Myc+sordd2, GMR-gla4 UAS-Xbp1 ATF4-mCherry UAS-Rh1$^{G69D}$–Myc/+;UAS-sordd2/+*) reduced Rh1$^{G69D}$–Myc levels as well as Xbp1-EGFP and ATF4-mCherry signals. In contrast, expressing CG8141 (*Rh1$^{G69D}$–Myc+CG8141, GMR-gla4 UAS-Xbp1 ATF4-mCherry UAS-Rh1$^{G69D}$–Myc/+;UAS-CG8141/+*) did not affect levels of Rh1$^{G69D}$–Myc, Xbp1-EGFP, or ATF4-mCherry. The Rh1$^{G69D}$–Myc epitope is labeled white; spliced Xbp1-GFP is labeled green; mCherry under control of the ATF4 promoter/5' UTR is labeled red. Scale bar, 50 μm. (b) Quantification of relative fluorescence from a; error bars indicate SEM (n = 4); ns, not significant; $^{****}$p<0.0001 (one-way ANOVA, Sidak's multiple comparisons test). (c) Western blot analysis shows the level of Rh1$^{G69D}$–Myc is reduced by expressing SORDD1 and SORDD2 in pupa eyes. Note Rh1$^{G69D}$ can form a dimer. (d) Quantification of **b**, error bars indicate SEM; n = 3, $^{**}$p<0.01, $^{***}$p<0.001 (one-way ANOVA, Sidak's multiple comparisons test). (e) E3 ubiquitin ligase activity of SORDD1, SORDD2 and HRD1 is essential for suppression of Rh1$^{G69D}$ induced cell degeneration. SORDD1 (*sordd1$^{C165S}$*), SORDD2 (*sordd2$^{C165S}$*) and HRD1 (*hrd1$^{C327S}$*) are inactivated by mutating the conserved Cys on the active site of the RING domain to Ser. Scale bar, 100 μm. (f) Ubiquitination of Rh1$^{G69D}$ by SORDD1 *in vivo*. Head lysate of *Rh1$^{G69D}$–Myc* (*GMR-gla4 UAS-Rh1$^{G69D}$–Myc/rpn12$^{RNAi}$*) flies co-expressing SORDD1 (*GMR-gla4 UAS-Rh1$^{G69D}$–Myc/rpn12$^{RNAi}$;UAS-sordd1/+*) or SORDD1$^{C165S}$ (*GMR-gla4 UAS-Rh1$^{G69D}$–Myc/rpn12$^{RNAi}$;UAS-sordd1$^{C165S}$/+*) were immunoprecipitated with anti-Myc antibody, and stained against

Myc (left) or ubiquitin (right). The proteasomal degradation of Rh1$^{G69D}$ was inhibited by knocking down the proteasome component Rpn12 (*rpn12$^{RNAi}$*) [54]. (g) Quantification of relative ubiquitination levels of Rh1$^{G69D}$–Myc from f, error bars indicate SEM; n = 3, ns, not significant, ****p<0.0001 (one-way ANOVA, Sidak's multiple comparisons test).

Rh1$^{G69D}$-induced retinal degeneration. We expressed ~7,000 RNAi lines in flies that co-expressed SORDD1 and Rh1$^{G69D}$ and found 7 RNAi lines that reduced the effect of SORDD1 on Rh1$^{G69D}$-induced retinal degeneration, but did not cause eye damage when knocked down alone (Fig 3A and Table 1).

Knocking-down the proteasome component Rpn12, or the VCP components TER94/VCP or UFD1 (Ubiquitin Fusion-Degradation 1), which are required to extract substrate from the ER into the cytosol, caused eye damage in *GMR>Rh1$^{G69D}$ sordd1* flies, but did not cause eye damage when knocked down alone (Fig 3B). Moreover, disruption of both VCP and the proteasome increased Rh1$^{G69D}$ protein levels in *GMR>Rh1$^{G69D}$, sordd1* retinae (S3B and S3C Fig and S6 Data). Based on these results, we conclude that SORDD1-mediated degradation of Rh1$^{G69D}$ is dependent on the VCP/proteasome system.

Degrading misfolded ER membrane proteins, which is called ERAD-M, requires proteins to be retro-translocated into the cytosol and subsequently poly-ubiquitinated [28]. HRD1, in addition to functioning as an E3 ubiquitin ligase, also complexes with the ER luminal protein HRD3 to form a retro-translocation channel for the movement and degradation of misfolded ER proteins. [29, 30]. However, knocking down *hrd1* or *hrd3* did not affect the eyes of *GMR>Rh1$^{G69D}$ sordd1* flies, raising the possibility that SORDD1/2-dependent ERAD does not rely on retro-translocation of substrates by the HRD1/3 complex (Fig 3C). To confirm the dispensable role of the HRD1/3 complex in SORDD1-mediated protein degradation, we used the "*ey-flp/hid*" system to generated flies with eyes that were homozygous mutant for either *hrd1* or *hrd3* [16]. SORDD1 continued to suppress Rh1$^{G69D}$-induced retinal degeneration in the absence of HRD1 or HRD3 (Fig 3C and 3D and S7 Data). Taken together, we conclude that SORDD1-mediated Rh1$^{G69D}$ degradation depends on VCP and the proteasome, but is independent of the HRD1/3 complex.

## SORDD1/2 and Hrd1 function redundantly in promoting Rh1$^{G69D}$ degradation

Since the E3 ligases SORDD1/2 and HRD1 could independently reduce levels of Rh1$^{G69D}$ and mitigate associated cell damage when overexpressed, we next asked whether endogenous SORDD1 or Hrd1 are involved in degrading Rh1$^{G69D}$. We expressed GFP-tagged Rh1$^{G69D}$ under its own promoter (*ninaE, neither inactivation nor afterpotential E*), and generated a deletion mutation that lacked both the *sordd1* and *sordd2* genes (*sordd1/2$^1$*) using the CRISPR/Cas9 system (S4 Fig). Homozygous *sordd1/2$^1$* mutants were viable and levels of Rh1$^{G69D}$ were not altered in *sordd1/2$^1$* or *hrd1$^1$* flies. However, Rh1$^{G69D}$ protein accumulated in animals that lacked both *sordd1/2$^1$* and *hrd1$^1$* (Fig 4A and 4B and S8 Data), suggesting that SORDD1/2 and HRD1 function in two parallel ERAD pathways to degrade Rh1$^{G69D}$.

To further explore the physiological roles of SORDD1/2 and HRD1 in maintaining photoreceptor cell homeostasis, we examined photoreceptor cell integrity in *sordd1/2$^1$* and *hrd1$^1$* mutants that expressed wild-type rhodopsin. We first used deep pseudopupil analysis, which reflects the compact structure of photoreceptor cells, assess levels or retinal degeneration [31]. Animals lacking *sordd1/2$^1$* or *hrd1$^1$* had normal retinal cytoarchitecture across a broad range of ages, whereas double mutants lacking both *sordd1/2$^1$* and *hrd1$^1$* exhibited a gradual loss of the deep pseudopupil, indicating age-dependent retinal degeneration (Fig 4C and S9 Data).

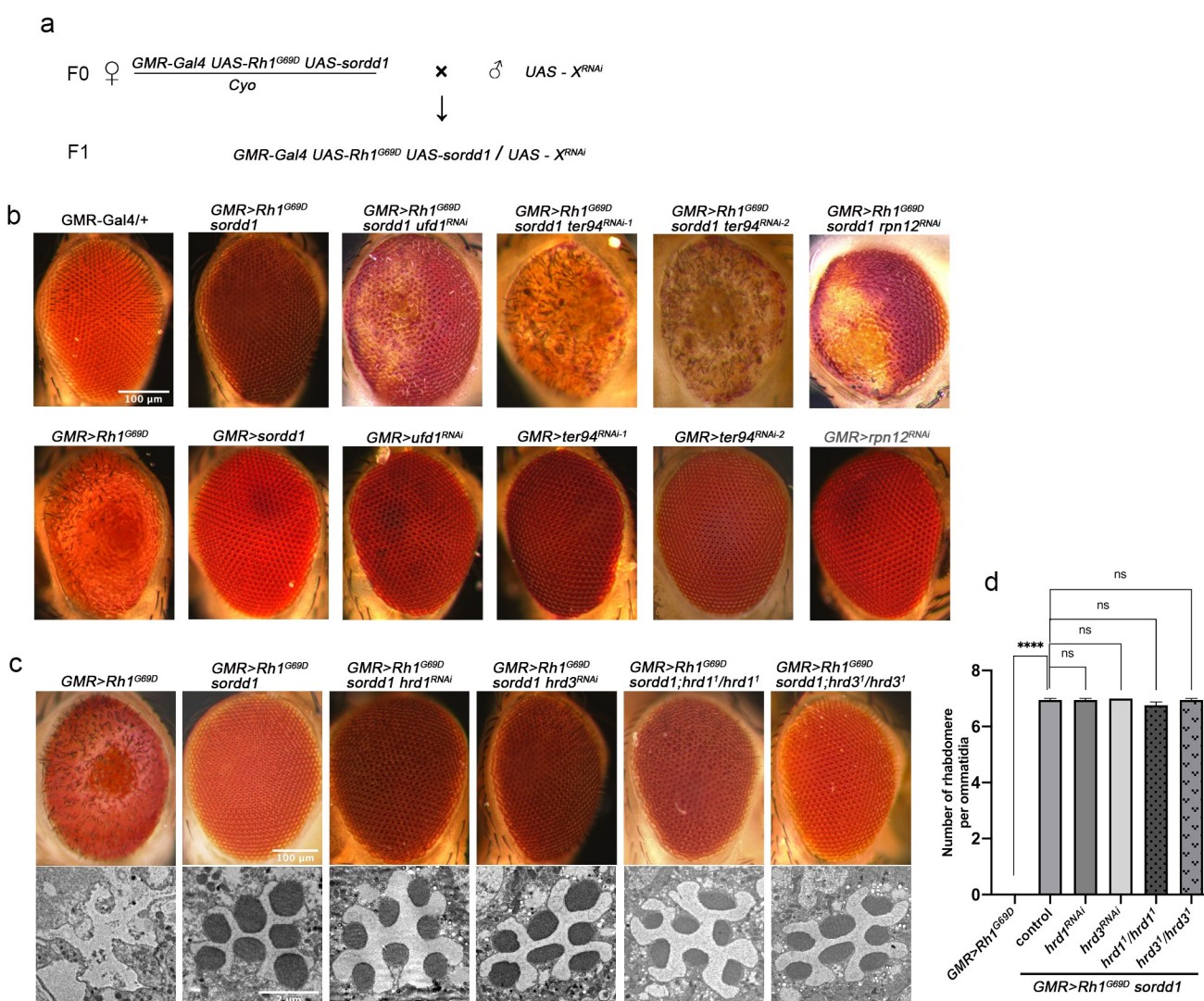

**Fig 3. Degradation of Rh1$^{G69D}$ by SORDD1 is dependent on VCP and proteasome, not dependent on the HRD1 complex.** (a) Schematic diagram of the genome-wide RNAi screen for modifiers of SORDD1-mediated Rh1$^{G69D}$ degradation. (b) Light photomicrographs show the adult eye was damaged when *ufd1* (*GMR>Rh1$^{G69D}$ sordd1 ufd1$^{RNAi}$*: GMR-Gal4 UAS-Rh-1$^{G69D}$ UAS-sordd1/+;UAS-ufd1$^{RNAi}$/+), *ter94* (*GMR>Rh1$^{G69D}$ sordd1 ter94$^{RNAi-1}$*: GMR-Gal4 UAS-Rh1$^{G69D}$ UAS-sordd1/+;UAS-ter94$^{RNAi-1}$/+ and *GMR>Rh1$^{G69D}$ sordd1 ter94$^{RNAi-2}$*: GMR-Gal4 UAS-Rh1$^{G69D}$ UAS-sordd1/+;UAS-ter94$^{RNAi-2}$/+) and *rpn12* (*GMR>Rh1$^{G69D}$ sordd1 rpn12$^{RNAi}$*: GMR-Gal4 UAS-Rh1$^{G69D}$ UAS-sordd1/+;UAS-rpn12$^{RNAi}$/+) were knocked down by *shRNA* in *GMR>Rh1$^{G69D}$ sordd1* background. Expressing *ufd1$^{RNAi}$* (*GMR>ufd1$^{RNAi}$*: GMR-Gal4 /+;UAS-ufd1$^{RNAi}$/+), *ter94$^{RNAi}$* (*GMR>ter94$^{RNAi}$*: GMR-Gal4 /+; UAS-ter94$^{RNAi}$/+) and *rpn12$^{RNAi}$* (*GMR>rpn12$^{RNAi}$*: GMR-Gal4 /+;UAS-rpn12$^{RNAi}$/+) alone have no obvious phenotype. Scale bar, 100 μm. (c) Mutations in *hrd1* or *hrd3* had no effect on eye morphology of *GMR>Rh1$^{G69D}$*, *sordd1* flies, both under light microscopy (top) and transmission electron microscopy (bottom). Homozygous null mutants of *hrd1* or *hrd3* were generated using "*ey-flp/hid*" system. Genotypes: *GMR>Rh1$^{G69D}$ sordd1 hrd1$^{RNAi}$* (GMR-Gal4 UAS-Rh-1$^{G69D}$ UAS-sordd1/+;UAS-hrd1$^{RNAi}$/+), *GMR>Rh1$^{G69D}$ sordd1;hrd1$^1$/hrd1$^1$*(ey-flp;GMR-Gal4 UAS-Rh1$^{G69D}$ UAS-sordd1/+; FRT82B hrd1$^1$/ FRT82B GMR-hid CL B), *GMR>Rh1$^{G69D}$ sordd1 hrd3$^{RNAi}$* (GMR-Gal4 UAS-Rh-1$^{G69D}$ UAS-sordd1/+;UAS-hrd3$^{RNAi}$/+) and *GMR>Rh1$^{G69D}$ sordd1;hrd3$^1$/ hrd3$^1$*(ey-flp;GMR-Gal4 UAS-Rh1$^{G69D}$ UAS-sordd1/+;FRT82B hrd3$^1$ /FRT82B GMR-hid CL). LM scale bar, 100 μm, TEM scale bar, 2 μm. (d) Quantification of rhabdomere numbers per ommatidia in **c**; error bars indicate SEM; sections from three flies of each genotype were used for quantification; ns, not significant (one-way ANOVA, Sidak's multiple comparisons test).

The rhabdomere is a microvilli structure that is functionally equivalent to the mammalian rod and cone outer segments. In wild-type ommatidia, all seven photoreceptor cells have intact rhabdomeres. To assay the detailed ultrastructure of photoreceptors in *sordd1/2$^1$* and *hrd1$^1$* mutants we used transmission electronic microscopy (TEM) (Fig 4D). Consistent with the pseudopupil analysis, both *sordd1/2$^1$* and *hrd1$^1$* mutant flies contained intact rhabdomeres and

**Table 1. List of genes identified in RNAi screen for factors required for SORDD1's function.**

| Stock NO. | Gene name | CG NO. | GO annotations |
|---|---|---|---|
| THU4245 | *ufd1* | CG6233 | Ubiquitin ligase binding |
| THU3262 | *ter94* | CG2331 | ATPase activity |
| THU3613 | *rpn12* | CG4157 | Ubiquitin-dependent protein degradation |
| THU2721 | *prsosα1* | CG18495 | Endopeptidase activity |
| TH03187.N | *tsf1* | CG6186 | Metal ion binding |
| TH04520.N | *CG9962* | CG9962 | Protein serine/threonine kinase activity |
| THU0769 | *amalgam* | CG2198 | - |

photoreceptor cells at 20 days of age, whereas 20-day-old *sordd1/2$^1$;hrd1$^1$* flies exhibited severe retinal degeneration with prominent vacuoles and obvious loss of rhabdomeres and photoreceptors. To verify that age-dependent retinal degeneration in *sordd1/2$^1$;hrd1$^1$* mutants results from the disruption of Rh1 homeostasis, we examined levels of Rh1 protein as well as levels of other phototransduction-related proteins such as TRP and INAD. In 3-day-old flies, both TRP and INAD levels were unaffected in *sordd1/2$^1$*, *hrd1$^1$* and *sordd1/2$^1$;hrd1$^1$* flies (compared with controls), indicating no retinal degeneration. However, Rh1 levels were significantly reduced in *sordd1/2$^1$;hrd1$^1$* mutants, suggesting that disruption of Rh1 homeostasis may lead to age-dependent retinal degeneration in flies lacking both SORDD1/2 and HRD1(Fig 4E, 4F and S10 Data). These results demonstrate that SORDD1/2 and HRD1 play redundant roles in maintaining rhodopsin homeostasis in photoreceptor cells, and that loss of both can lead to retinal degeneration.

## SORDD1 strongly suppresses retinal degeneration in the *ninaE$^{G69D}$* model of adRP

To validate that SORDD1/2 is a *bona fide* suppressor of adRP, we used a classic model in which the dominant *ninaE$^{G69D}$* mutation leads to age-dependent retinal degeneration [20, 21]. The *ninaE$^{G69D}$* heterozygous flies (*ninaE$^{G69D}$*/+) exhibit severe loss of rhabdomeres and photoreceptor cells ~25 days after eclosion (Fig 5A, 5B, 5E and S11 Data). Expression of Hrd1 (*ninaE$^{G69D}$*/ *GMR-hrd1*) in these eyes improved photoreceptor cell survival (Fig 5C–5E), and expression of SORDD1 (*ninaE$^{G69D}$*/*GMR-sordd1*) completely rescued the phenotype (Fig 5D and 5E).

We examined the function of these photoreceptor neurons using an electroretinogram (ERG) assay, which is an extracellular recording that measures the summed light responses of photoreceptor cells. The ERG exhibits two primary features: a rapid corneal negative response, reflecting phototransduction; and on- and off-transients, which reflect synaptic transmission from photoreceptor cells [32]. Twenty-five-day-old *ninaE$^{G69D}$*/+ flies exhibited small ERG response amplitudes and a complete loss of transients, suggesting disruption of several neural functions (Fig 5F and 5G and S12 Data). Consistent with the results obtained via TEM, *GMR-hrd1; ninaE$^{G69D}$*/+ flies showed significant increased ERG amplitude at the age of 25 days relative to *ninaE$^{G69D}$*/+ flies, and expression of SORDD1 fully restored the ERG amplitude and on/off transients, indicating a complete rescue of photoreceptor function (Fig 5F and 5G and S12 Data). Furthermore, at 32 days, while *ninaE$^{G69D}$* flies had no ERG response, ERG responses and transients were detected in *ninaE$^{G69D}$*/*GMR-sordd1* flies (Fig 5F).

## Discussion

This work suggests a new ERAD-M pathway that depends on the E3 ubiquitin ligase SORDD1/2 (Fig 4G). Misfolded membrane proteins such as Rh1$^{G69D}$ may be ubiquitinated by

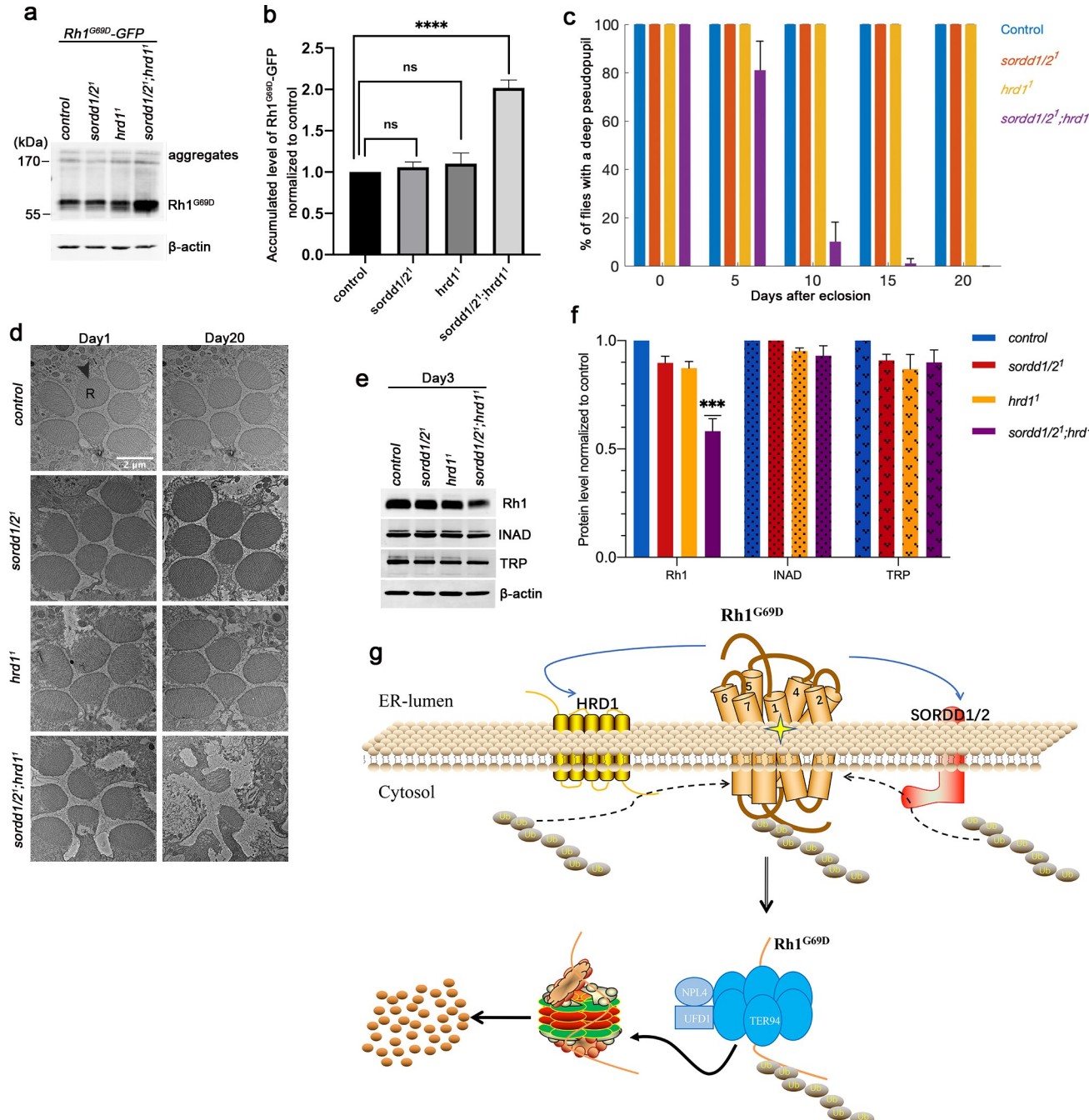

**Fig 4. SORDD1/2 and Hrd1 function redundantly in promoting Rh1$^{G69D}$ degradation.** (a) Western blot shows Rh1$^{G69D}$-GFP accumulation in *sordd1/2$^1$; hrd1$^1$* double mutants (*ey-flp sordd1/2$^1$;FRT82B ninaE-Rh1$^{G69D}$-GFP hrd1$^1$/FRT82B GMR-hid CL*), compared to *sordd1/2$^1$* (*sordd1/2$^1$;ninaE-Rh1$^{G69D}$-GFP*) and *hrd1$^1$* (*ey-flp;FRT82B ninaE-Rh1$^{G69D}$-GFP hrd1$^1$/FRT82B GMR-hid CL*) mutant alone. Rh1$^{G69D}$-GFP is driven by an endogenous *ninaE* promoter. The higher molecular weight of Rh1$^{G69D}$ aggregates could be seen at the top of the lanes. (b) Quantification of **a**, error bars indicate SEM; n = 3, ns, not significant, ****p<0.0001 (ANOVA, Sidak's multiple comparisons test). (c) Quantification of the percentage of the deep pseudopupil in control (*ey-flp;FRT82B*), *sordd1/2$^1$*, *hrd1$^1$* (*ey-flp;FRT82B hrd1$^1$/FRT82B GMR-hid CL*) and *sordd1/2$^1$;hrd1$^1$* (*ey-flp sordd1/2$^1$;FRT82B hrd1$^1$/FRT82B GMR-hid CL*) flies at 0, 5, 10, 15 and 20 days old. At least ~100 flies in 3 groups were measured for each genotype. Error bars indicate SEM. (d) TEM images of 1-day-old and 20-day-old adult eye tangential sections. Genotypes are as indicated on the left side of the panels. R, rhabdomere. Scale bar, 2 μm. The *sordd1/2$^1$;hrd1$^1$* double mutant flies show severe degeneration of photoreceptor cells at age of 20 days. All flies are in white eye background, and raised under 12 h light /12 h dark cycles. (e) Western blot analysis of Rh1, TRP, and INAD in control, *sordd1/2$^1$*, *hrd1$^1$*, and *sordd1/2$^1$;hrd1$^1$* flies at 3 days old. (f) Quantification of relative protein levels of Rh1, TRP, and INAD in **e**; error bars indicate SEM (n = 3); ***p<0.001 (one-way ANOVA, Sidak's multiple comparisons test). (g) Proposed model of SORDD1/2 in the degradation of misfolded Rh1$^{G69D}$ parallel to HRD1 complex.

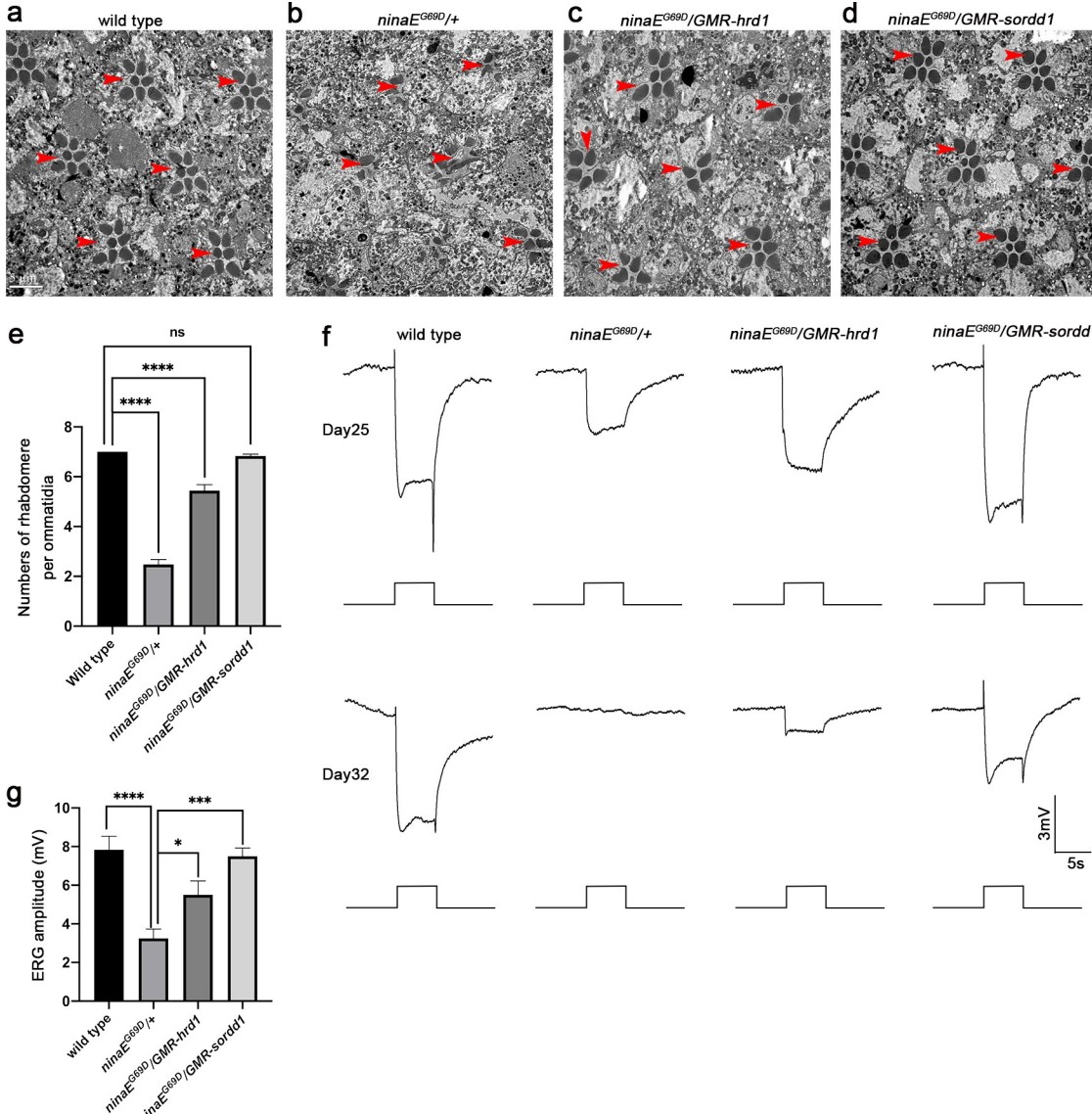

**Fig 5. SORDD1 strongly suppresses retinal degeneration in the *ninaE^G69D* model of adRP.** (a-d) Representative TEM images of 25-day-old adult eye tangential sections of (a) wild type (*canton S*), (b) *ninaE^G69D/+*, (c) *ninaE^G69D/GMR-hrd1* and (d) *ninaE^G69D/GMR-sordd1* flies. Rhabdomeres are indicated by red arrows. Scale bar, 5 μm. (e) Quantification of **a-d**, Error bars indicate SEM; sections from three individual flies of each genotype were used for quantification. ns, not significant, \*\*\*\*p<0.0001 (one-way ANOVA, Sidak's multiple comparisons test). (f) ERG recordings of 25-day-old and 32-day-old wild-type, *ninaE^G69D/+*, *ninaE^G69D/GMR-hrd1* and *ninaE^G69D/GMR-sordd1* flies. Flies were exposed to a 5-s pulse of orange light after 1 min of dark adaptation. (g) Statistic results of the amplitude of ERG recordings of flies at age of 25 days, Error bars indicate SEM; n = 6, \*p<0.1, \*\*\*p<0.001, \*\*\*\*p<0.0001 (one-way ANOVA, Sidak's multiple comparisons test). Red-eyed flies were raised under constant light conditions. All flies were in *w^+* background and were raised in 24 h constant light (~700 Lux) at 25°C.

two independent ER enzymes, HRD1 and SORDD1/2. The ubiquitylated protein is then transported out of the ER membrane via the VCP complex and degradation in cytosolic proteasomes. Although HRD1 also functions (in complex with HRD3) as a retro-translocation channel to facilitate removal of substrates from the ER [29, 30], HRD1 and HRD3 are dispensable for SORDD1-mediated degradation of Rh1^G69D; thus, SORDD1 suppresses Rh1^G69D-induced ER stress and cell damage independent of HRD1 and HRD3. Although we

demonstrated that VCP complex components are essential for SORDD1-mediated degradation of Rh1$^{G69D}$, we did not identify a candidate retro-translocation channel. One possibility is that Rh1$^{G69D}$ ubiquitinated by SORDD1 is removed directly from the ER membrane by the VCP complex without channels due to the plasticity of the ER membrane [33]. Alternatively, key translocation channels, such as Sec61, may serve dual functional roles and be essential for cell viability [34].

Disrupting rhodopsin homeostasis–the most abundant protein in photoreceptors–induces severe stress and frequently results in the degeneration of photoreceptor neurons [35]. Degradation of misfolded rhodopsin via ERAD is one of several mechanisms that maintains rhodopsin homeostasis and normal photoreceptor function and viability. Despite HRD1 serving as a key E3 ligase in ERAD, null *hrd1* mutants exhibit normal rhodopsin levels and light responses [16, 19]. One possibility is that, in addition to being degraded by ERAD, misfolded rhodopsin might also be targeted by autophagy pathways [36, 37]. However, recent reports argue against a complementary function of autophagy in degrading misfolded rhodopsin–inhibiting autophagy reduced retinal degeneration in *Rho*$^{P23H}$ mice by promoting proteasomal degradation of misfolded mutant rhodopsin [17, 18, 38].

Here we present data suggesting that SORDD1/2 functions in a similar, and independent, manner to degrade mutant Rh1$^{G69D}$. While knocking out either *hrd1* or *sordd1/2* did not prevent the degradation of Rh1$^{G69D}$ protein, Rh1$^{G69D}$ accumulated in photoreceptors that lacked both E3 ubiquitin ligases. Moreover, flies lacking both *hrd1* and *sordd1/2* exhibit severe retinal degeneration, whereas photoreceptor cells remain intact in flies lacking either *hrd1* or *sordd1/2*. Therefore, SORDD1 functions redundantly with HRD1 in ERAD of misfolded rhodopsin, thereby helping to maintain rhodopsin homeostasis. Two redundant ERAD pathways might be beneficial for the maximum clearance of misfolded rhodopsin and to maintain photoreceptor cell function.

Because increasing the efficiency of protein folding in the ER by expressing BiP prevents retinal degeneration in Rho$^{P23H}$ models, chemical chaperones capable of reducing Rho misfolding are promising therapeutic interventions [5, 39–41]. However, the escape of mutant forms of rhodopsin from the ER might contribute to defects in the morphogenesis of rod photoreceptor cell outer segments [42]. Recently, it was shown that increasing proteasome-mediated degradation of misfolded rhodopsin protects against adRP [17, 18, 38]. However, induction of proteasomal activity might be problematic, since it impairs general cellular proteostasis. In a fly model of adRP, inhibition of VCP or the proteasome alleviated retinal degeneration caused by Rh1$^{P37H}$, despite the fact the clearance of misfolded RH1$^{P37H}$ was disrupted [24]. This might be because inhibition of the VCP or the proteasome may suppress a cell death pathway that acts downstream of misfolded rhodopsin. Indeed, VCP is required for autophagy flux, and blocking autophagy suppresses retinal degeneration in *Rho*$^{P23H}$ mice [38, 43, 44]. In flies co-expressing SORDD1 and Rh1$^{G69D}$, disruption of VCP or the proteasome dramatically blocked SORDD1-accelerated degradation of Rh1$^{G69D}$, thus reversing the suppression of Rh1$^{G69D}$-induced eye damage by SORDD1. Given that E3 ubiquitin ligases are the rate-limiting step of ERAD, inducing ERAD by overexpression of HRD1 effectively reduces levels of misfolded Rh1$^{G69D}$ protein and suppresses retinal degeneration in this fly model of adRP. This raises the possibility that key ERAD components may be a promising therapeutic target for treating adRP. However, because ERAD regulates the turn-over of a variety of very important proteins [40, 45, 46], *hrd1* mutations are lethal, as is the widespread overexpression of HRD1 (e.g., via the *tubulin* or *actin* promoter).

Unlike HRD1, flies that lack both *sordd1* and *sordd2*, or flies that express both SORDD1 and SORDD2 ubiquitously, do not exhibit obvious defects, suggesting that SORDD1 is more specific to misfolded proteins. Moreover, overexpression of SORDD1 efficiently degraded

Rh1$^{G69D}$ and, therefore, better suppressed retinal degeneration and restored light response than HRD1. Furthermore, ERAD of misfolded proteins via SORDD1 is likely conserved in mammals, as RNF5 and RNF185 also suppressed Rh1$^{G69D}$-induced retinal degeneration. Indeed, both E3 ligases have been implicated in ERAD of a misfolded version of the cystic fibrosis transmembrane conductance regulator, CFTR$^{\Delta F508}$, which is associated with cystic fibrosis [25, 47]. However, unlike SORDD1, which suppressed Rh1$^{G69D}$–associated retinal degeneration, genetic and pharmacological inhibition of RNF5 *in vivo* attenuates intestinal pathological phenotypes in a mouse model of cystic fibrosis without blocking degradation of CFTR$^{\Delta F508}$ [48, 49]. This might be because RNF5 and RNF185 function redundantly to degrade CFTR$^{\Delta F508}$ [25]. Moreover, RNF185 has recently been shown to target ER membrane proteins, including CYP51A1 and TMUB2 [26]. Since SORDD1 is ubiquitously expressed, it might also be involved in the degradation and turnover of a subset of membrane proteins. All the evidence strongly suggests that SORDD1 and its human homologs play roles in the quality control of membrane proteins. Overall, these data suggest that the facilitation of SORDD1-dependent ERAD of misfolded rhodopsin is a promising treatment for adRP.

## Materials and methods

### Fly stocks

*UAS–xbp1–EGFP* and *UAS–hrd1* have been reported previously [19, 50]. The flyORF collection and EP collection for screening suppressors of Rh1$^{G69D}$ induced cell degeneration were obtained from the Zurich ORFeome Project (https://www.flyorf.ch) and Bloomington Drosophila Stock Center (https://bdsc.indiana.edu), respectively. The *hrd3$^1$* mutant flies (*P[RS3] hrd3$^{CB-0952-3}$*) and *ninaE$^{G69D}$* flies were obtained from Bloomington Drosophila Stock Center. The transgenic RNAi lines for the *in vivo* RNAi screen were obtained from TsingHua Fly Center (http://fly.redbux.cn). The RNAi lines used in this work are as follows: *P[TRiP. GL01251] attP2* (*ufd1$^{RNAi}$*), *P[TRiP. JF03402]attP2* (*ter94$^{RNAi-1}$*), *P[GD9777]v24354*(*ter94$^{RNAi-2}$*)*P[TRiP. HMS01032]attP2* (*rpn12$^{RNAi}$*), *P[TRiP. JF02711]attP2* (*prosα1$^{RNAi}$*), *P[TRiP. HMS00297]attP2* (*ama$^{RNAi}$*). The *hrd1$^1$ FRT82B/TM3* flies was reported previously [16]. The *ey-flp; FRT82B GMR-hid CL/TM3* flies were maintained in the laboratory of T. Wang. All flies were maintained under 12 h light /12 h dark cycles at 25°C unless mentioned.

### Generation of transgenic flies

The cDNA sequences of *Rh1*, *sordd1*, *sordd2*, *hrd1* and *CG32847* were amplified from RH01460, GH14055, RE35552, GH11117 and FI06431 of DGRC gold cDNA collections, respectively (*Drosophila* Genomics Resource Center). cDNA of *CG8141* was amplified from the RT-PCR products of *w$^{1118}$* flies. The cDNA sequences of human genes *HsSYVN1*, *HsRNF5* and *HsRNF185* were amplified from ORF Clones IOH21699, IOH3743 and IOH14506 obtained from Thermo Fisher Scientific. These cDNA sequences were then subcloned into the *pUAST-attB* or *GMR-attB* vectors. *Rh1$^{G69D}$* was mutated from the *ninaE* cDNA and subcloned into the *pninaE-attB* vector with a c-terminal GFP-tag. The constructs were injected into *M (vas-int.Dm)ZH-2A;M(3xP3-RFP.attP)ZH-86Fb* or *M(vas-int.Dm)ZH-2A;M(3xP3-RFP.attP) ZH-51C* embryos, and transformants were identified based on eye color. The *3XP3-RFP* markers were eliminated by crossing to a Cre-expressing line.

To generate *UAS-Rh1$^{G69D}$* or *UAS-Rh1$^{G69D}$-Myc* flies, the *Rh1$^{G69D}$* cDNA was subcloned to *pUAST* vector with or without Myc tag, and the constructs were injected into *w$^{1118}$* embryos and transformants with random insertions were identified based on eye color. The 2$^{nd}$ chromosome insertions with weak *Rh1$^{G69D}$* expressing were used for experiments.

To generated the *ATF4-mCherry* reporter flies, the genomic DNA sequence with endogenous *ATF4* promoter sequence (−906 bp upstream of the transcription starting site) and complete 5'UTR including starting ATG (1102 bp from the transcription starting site to the translation starting ATG) were fused with mCherry cDNA and replaced the *UAS* sequence of the *pUAST-attB* vector. The *ATF4-mCherry* constructs were injected into embryos of *M(vas-int.Dm)ZH-2A;M(3xP3-RFP.attP)ZH-51C* and *M(vas-int.Dm)ZH-2A;M(3xP3-RFP.attP)ZH-86Fb*, and transformants with eye colors were crossed with a Cre-expressing line to remove 3XP3-RFP and mini-white markers.

## Generation of *sordd1 and sordd2* knockout flies (*sordd1/2*[1])

The *sordd1/2*[1] mutations were generated by the Cas9/sgRNA system [51]. Two recognition sequences of guiding RNA to the *sordd1* and *sordd2* locus were designed (sgRNA1: 5'-GAGCGTGGACTTTGGTCCGG-3'; sgRNA2: 5'-AATACGAAAACAGCTTGGAC -3') and cloned into the *U6b-sgRNA-short* vector. The plasmids were injected into the embryos of nos-Cas9 flies. The successful *sordd1/2* deletion mutants (*sordd1/2*[1]) were identified screened by PCR with a product of ~800 bp using a pair of primers (5'-GAAAACATCCTCTAGTTCGC-3' and 5'- ATCAATCGAGGACTGATCACTGAT-3') (S4B Fig). The transcription of *sordd1* and *sordd2* was completely absent in *sordd1/2*[1] mutants by RT-PCR using a pair of primers (5'-ATGGAGGAACCGAAAAGTGTAC-3' and 5'- CTATGCATAGAACAGCCACAAT-3') (S4C Fig).

## RNA extraction and quantitative real-time PCR analysis

Larvae or adult flies were dissected to obtain the different tissues and tissue-specific RNA was extracted using TRIzol Reagent (Invitrogen) according to the manufacturer's protocol. Total RNA was reverse-transcribed using PrimeScript RT-PCR kit (Takara). qRT-PCR was performed using SYBR Premix Ex Taq (Takara) on a CFX96 real time PCR detection system (Bio-Rad). The average threshold cycle value (CT) was calculated from at least three replicates per sample. Expression of sordd1 and sordd2 was standardized relative to rp49.Relative expression values were determined by the △△CT method. Primers:

*sordd1*-f: CTAATCGGGGAACAAGGCAATACG, *sordd1*-r: CTATGCATAGAACAGCCACAATAT;

*sordd2*-f: ATATTAGAAGGCTACACGAAGATG, *sordd2*-r: TTATCGGGATGATCCGTGCACTAT;

*rp49*-f: CAGTCGGATCGATATGCTAAGCTG, *rp49*-r: TAACCGATGTTGGGCATCAGATAC

## Immunofluorescence staining

Third instar larva imaginal eye discs were dissected and fixed in PBS with 4% paraformaldehyde (Sigma) for 1h at room temperature, followed by incubation with primary antibodies including mouse anti-Myc (1:200; Santa Cruz), rabbit anti–GFP (1:200; Invitrogen), rat anti–RFP antibody (1:200; Chromotek) overnight at 4°C. Then incubated samples were washed in PBST (0.3% Triton X-100) buffer 5 times. Finally, samples were incubated with Alexa Fluor 488–conjugated, Alexa Fluor 568–conjugated, Alexa Fluor 647–conjugated secondary antibodies (1:500; Invitrogen), and 20 nM DAPI (Invitrogen) for 1 h at room temperature. Fluorescence images were acquired at room temperature with an A1 confocal laser scanning microscope (Nikon) using a 20x objective (Plan Fluor NA 1.30; Nikon) and an A1+ camera (Nikon).

To examine the cellular localization of SORDD1, S2 cells were seeded onto coverslips coated with concanavalin A and transformed with Myc-tagged SORDD1 plasmid. After 36 h of transformation, the cells were fixed using 4% paraformaldehyde in PBS for 0.5 h, followed by incubation with primary antibodies including mouse anti-Myc (1:200; Santa Cruz) and anti-Calnexin (1:100; Developmental Studies Hybridoma Bank) for 1.5 h at room temperature. Samples were incubated with Alexa Fluor 488–conjugated, Alexa Fluor 568–conjugated secondary antibodies (1:500; Invitrogen) with 20 nM DAPI (Invitrogen) for 1 h at room temperature. Fluorescence images were acquired at room temperature with an A1 confocal laser scanning microscope (Nikon) using a 60x objective.

## Fractionation assay

Fractionation assays were performed with a Membrane and Cytosol Protein Extraction Kit (Beyotime Biotechnology) according to the manufacturer's protocol. Briefly, S2 cells were grown in 6 cm dishes and transformed with Myc-tagged SORDD1 plasmid. After 48 h of transformation, the cells were harvested and lysed with Buffer A, followed by centrifugation at 14,000g for 30 min at 4°C. The supernatant (S) was collected, and the pellet was lysed again by Buffer B, followed by centrifugation at 14,000g for 5 min at 4°C to collect the pellet (P). S and P fractions were analyzed by western blot.

## Immunoprecipitation and western blots

Mashed fly heads were lysed with 10 mM Tris-HCl lysis buffer (pH 7.4, 150 mM NaCl, 0.5 mM EDTA, 0.5% NP-40 with 1× proteinase inhibitor cocktail [Roche]) for 30 min, and centrifuged at 16,100 g. The supernatant was used for subsequent immunoprecipitation with anti-Myc beads (Chromotek). Beads were washed in TBS buffer with NP-40 (10 mM Tris–Cl (pH7.4), 150 mM NaCl, 0.5 mM EDTA, and 50 nM NP-40 with 1× proteinase inhibitor cocktail) three times, and boiled in SDS loading buffer for standard western blot assays.

For western blot assays, dissected pupa eyes or adult fly heads were homogenized in SDS loading buffer for SDS-PAGE. The blots were probed with primary antibodies against Myc (rabbit, 1:1000; Santa Cruz), mouse anti-β-actin (1:2000; Santa Cruz), mouse anti-ubiquitin (1:1000; Santa Cruz), rabbit anti-GFP (1:2000; Origene), followed by incubation with IRDye 680 goat anti-mouse IgG (1:10000, LI-COR Biosciences) and IRDye 800 goat anti-rabbit IgG (1:10000, LI-COR Biosciences). Signals were detected using an Odyssey infrared imaging system (LI-COR Biosciences).

## Scanning electron microscopy

Dissected adult fly eyes were fixed in 2.5% glutaraldehyde at 4°C overnight followed by incubation in 1% osmium tetroxide for 1 h at room temperature. Samples were dehydrated in a series of ethanol dilutions (20, 35, 50, 70, and 100% ethanol), and substituted by liquid carbon dioxide. After the liquid carbon dioxide gasified, samples were scanned using a ZEISS EVOL LS10 scanning electron microscope at room temperature. The number of ommatidia in the middle 3 rows in each sample was counted to assess the damage degree, and images from 5 flies were used for quantification.

## Transmission electron microscopy

TEM was performed with standard methods as described [52]. Dissected adult fly eyes were fixed in 4% paraformaldehyde and 2.5% glutaraldehyde at 4°C overnight followed by incubation in 1% osmium tetroxide for 1 h at room temperature. Then samples were dehydrated in a

series of ethanol dilutions (20, 35, 50, 70, 80, 90, and 100% ethanol) and embedded in LR White resin (Polysciences, Inc.). Thin sections (80 nm) were stained with 0.06% uranyl acetate and 0.1% lead citrate (Sigma, St. Louis, MO) and examined using a JEM-1400 transmission electron microscope (JEOL) at room temperature. The images were acquired using a Gatan camera (model 794; Gatan, Inc.).

### Electroretinogram recordings

Electroretinogram (ERG) recordings were performed as described [53]. Microelectrodes filled with Ringer's solution were placed onto the surface of the compound eye and the thorax of a fixed fly. A Newport light projector (model 765) was used for stimulation. After 1 min of dark adaptation, red-eyed flies were exposed to a 5-s pulse of ~2000 lux orange light (source light was filtered using a FSR-OG550 filter, Newport). ERG signals were amplified with a Warner electrometer IE-210 and recorded with a MacLab/4 s analogue-to-digital converter and the clampelx10.2 program (Warner Instruments, Hamden, USA). All the flies used in ERG assay were raised under constant white light of ~700 Lux at 25°C.

### Statistics

Statistical results were generated by GraphPad Prism 8 and statistical significance was assessed through Ordinary one-way ANOVA, Sidak's multiple comparisons test analyses. All error bars represent S.E.M.

### Supporting information

**S1 Fig. SORDD1 and SORDD2 suppress Rh1$^{G69D}$-induced retinal degeneration.** (a-j) Scanning electron microscopy images show eye morphology of (a) control (*GMR-gal4/+*), (b) *GMR>Rh1$^{G69D}$*, (c) *GMR>Rh1$^{G69D}$+hrd1*, (d) *GMR>Rh1$^{G69D}$+sordd1*, (e) *GMR>Rh1$^{G69D}$+ sordd2*, (f) *GMR>Rh1$^{G69D}$+CG8141*, (g) *GMR>Rh1$^{G69D}$+ CG32847*, (h) *GMR>Rh1$^{G69D}$+ SYVN1*, (i) *GMR>Rh1$^{G69D}$+RNF5* and (j) *GMR>Rh1$^{G69D}$+RNF185*. Scale bar, 100 μm. (k) Quantification of the number of ommatidia in the middle 3 rows of **a-j** (relevant rows are indicated in **a**). Error bars indicate SEM (n = 5); ns, not significant; ****p<0.0001 (one-way ANOVA, Sidak's multiple comparisons test).
(TIF)

**S2 Fig. SORDD1 is an ER-associated E3 ligase.** (a) Protein sequence alignment between SORDD1 and SORDD2 showed they share 92.4% amino acid identity, and the differences mainly reside in the C terminus near the transmembrane region. Both of SORDD1 and SORDD2 are predicted to have a typical C3HC4 RING domain and position C165 is predicted to be the active center. (b)RNA sequencing and qPCR results of *w$^{1118}$* flies show the expression of sordd1 and sordd2 in different tissues. (c) Fractionation assay to show SORDD1 is a membrane protein. A Myc tag was fused to the N terminus of SORDD1 and S2 cells were transiently transfected with Myc-tagged SORDD1.S, supernatant. P, pellet. (d) Confocal images show the cellular localization of SORDD1.S2 cells were transiently transfected with Myc-tagged SORDD1 (red), and labeled with ER marker Calnexin (green) and DAPI (blue). Scale bar, 10 μm. (e) SORDD1/2 also suppresses retinal degeneration induced by earlier expression of wild-type Rh1 and Rh1$^{P37H}$. Light photomicrographs show adult eye morphology of control (*GMR-Gal4/+*), and flies overexpressing wild-type Rh1 (*GMR-Gal4/UAS-Rh1*) or Rh1$^{P37H}$ (*GMR-Gal4/UAS-Rh1$^{P37H}$*), together with SORDD1 (*GMR-Gal4/UAS-Rh1;UAS-sordd1/+* and *GMR-Gal4/UAS-Rh1$^{P37H}$;UAS-sordd1/+*) or SORDD2 (*GMR-Gal4/UAS-Rh1;UAS-sordd2/+*

and *GMR-Gal4/UAS-Rh1$^{P37H}$;UAS-sordd2/+*). Scale bar, 100 μm.
(TIF)

**S3 Fig. Ubiquitination sites of Rh1$^{G69D}$ are essential for suppression of Rh1$^{G69D}$ induced cell degeneration by SORDD1/2 and HRD1.** Potential ubiquitination sites of Rh1$^{G69D}$ are mutated by substitution of all 15 cytosol lysines and 2 luminal lysines with arginines (Rh1$^{G69D+KR}$). (a) the adult eye morphology of wild type (*GMR-gal4/+*) and flies overexpressing Rh1$^{G69D}$ (*GMR-Gal4/UAS-Rh1$^{G69D}$*), Rh1$^{G69D+KR}$ (*GMR-Gal4/UAS-Rh1$^{G69D+KR}$*) together with *sordd1* (*GMR-Gal4/UAS-Rh1$^{G69D+KR}$;UAS-sordd1/+*), *sordd2* (*GMR-Gal4/UAS-Rh1$^{G69D+KR}$; UAS-sordd2/+*) or *hrd1* (*GMR-Gal4/UAS-Rh1$^{G69D+KR}$;UAS-hrd1/+*). Scale bar, 100 μm. (b) Blocking VCP and proteasome components lead to the accumulation of Rh1$^{G69D}$ when SORDD1 are expressed. Western blot analysis of head extracts from 1-day-old flies shows the levels of Rh1$^{G69D}$–Myc in *GMR>Rh1$^{G69D}$-Myc sordd1* (*GMR-Gal4 UAS-Rh1$^{G69D}$-Myc UAS-sordd1/+*) flies increase by knocking down proteasome subunits Prosα1 (*prosα1$^{RNAi}$, GMR-Gal4 UAS-Rh1$^{G69D}$-Myc UAS-sordd1/+;UAS-prosα1$^{RNAi}$/+*) or Rpn12 (*rpn12$^{RNAi}$, GMR-Gal4 UAS-Rh1$^{G69D}$-Myc UAS-sordd1/+;UAS-rpn12$^{RNAi}$/+*), component of VCP complex UFD1 (*ufd1$^{RNAi}$, GMR-Gal4 UAS-Rh1$^{G69D}$-Myc UAS-sordd1/+;UAS-ufd1$^{RNAi}$/+*) or Amalgam (Ama), one of the genes identified in the *RNAi* screen (*ama$^{RNAi}$, GMR-Gal4 UAS-Rh1$^{G69D}$-Myc UAS-sordd1/+;UAS-ama$^{RNAi}$/+*). Note Rh1$^{G69D}$ can form dimer. (c) Statistic results of **b**, Error bars indicate SEM; n = 3, *p<0.1, ***p<0.001, ****p<0.0001 (one-way ANOVA, Sidak's multiple comparisons test).
(TIF)

**S4 Fig. Generation of *sordd1/2* deletion (*sordd1/2$^{1}$*) mutants.** (a) Scheme for generating the *sordd1/2* deletion by CRISPR-Cas9 system, the two sgRNA recognition sites and positions of the DNA primers used for PCR (arrows) are indicated. (b) PCR products of ~800 bp were obtained from *sordd1/2$^{1}$* mutants with successful deletion of *sordd1* and *sordd2* loci. (c) The verification of *sordd1/2$^{1}$* by RT-PCR.RT-PCR products of 834 bp were obtained from full-length *sordd1* cDNA in wild type while absent in *sordd1/2$^{1}$*.
(TIF)

**S1 Data. SEM quantification data.**
(XLSX)

**S2 Data. *sordd1* and *sordd2* mRNA expression level relative to *rp49* in different tissues.**
(XLSX)

**S3 Data. Larval eye discs fluorescence of Rh1$^{G69D}$, XBP1 and ATF4 in different genotypes.**
(XLSX)

**S4 Data. Pupa eye western blots data.**
(XLSX)

**S5 Data. *In vivo* ubiquitination assay data.**
(XLSX)

**S6 Data. Relative Rh1$^{G69D}$ protein levels under different gene knockdown backgrounds.**
(XLSX)

**S7 Data. TEM quantification data of number of rhabdomere per ommatidia under *hrd1/ hrd3* knockdown or knockout backgrounds.**
(XLSX)

**S8 Data. Relative Rh1$^{G69D}$ protein levels under *sordd1/2$^1$* mutants, *hrd1$^1$* mutants and *sordd1/2$^1$;hrd1$^1$* mutants backgrounds.**
(XLSX)

**S9 Data. Deep pseudopupil analysis data.**
(XLSX)

**S10 Data. Relative Rh1, INAD, TRP protein levels under *sordd1/2$^1$* mutants, *hrd1$^1$* mutants and *sordd1/2$^1$;hrd1$^1$* mutants backgrounds.**
(XLSX)

**S11 Data. TEM quantification data of number of rhabdomere per ommatidia of ninaE$^{G69D}$ under *hrd1* or *sordd1* overexpression backgrounds.**
(XLSX)

**S12 Data. Quantification data of ERG amplitude of ninaE$^{G69D}$ under *hrd1* or *sordd1* overexpression backgrounds.**
(XLSX)

## Acknowledgments

We thank the Bloomington Stock Center, TsingHua Fly Center, Drosophila Genomic Resource Center, the Zurich ORFeome Project, Developmental Studies Hybridoma Bank and Dr. HD. Ryoo for stocks and reagents. We thank Y. Chen, Y. Wang, X. Liu, J. Wang, and Dr. Z. Huang for technological assistance. We thank Dr. Lin Yang from the Institute of Genetics and Developmental Biology for assistance with TEM. We thank Drs. D. O'Keefe and G. Murphy for editing the manuscript.

## Author Contributions

**Conceptualization:** Jaiwei Xu, Tao Wang.

**Data curation:** Jaiwei Xu, Tao Wang.

**Formal analysis:** Jaiwei Xu, Tao Wang.

**Funding acquisition:** Tao Wang.

**Investigation:** Jaiwei Xu, Haifang Zhao, Tao Wang.

**Methodology:** Jaiwei Xu, Haifang Zhao.

**Project administration:** Tao Wang.

**Resources:** Tao Wang.

**Supervision:** Tao Wang.

**Validation:** Jaiwei Xu, Haifang Zhao.

**Visualization:** Jaiwei Xu, Haifang Zhao, Tao Wang.

**Writing – original draft:** Jaiwei Xu.

**Writing – review & editing:** Tao Wang.

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
