## [Decision Letter · Decision Letter 0]

8 Jun 2020

Dear Dr Wang,

Thank you very much for submitting your Research Article entitled 'Suppression of retinal degeneration by two novel ERAD ubiquitin E3 ligases SORDD1/2 in Drosophila.' to PLOS Genetics. Your manuscript was fully evaluated at the editorial level and by three independent peer reviewers. The reviewers appreciated the attention to an important problem, but raised some substantial concerns about the current manuscript. Based on the reviews, we will not be able to accept this version of the manuscript, but we would be willing to review again a much-revised version. We cannot, of course, promise publication at that time.

If you decide to revise the manuscript for further consideration at PLOS Genetics, please aim to resubmit within the next 60 days, unless it will take extra time to address the concerns of the reviewers, in which case we would appreciate an expected resubmission date by email to plosgenetics@plos.org.

[LINK]

We are sorry that we cannot be more positive about your manuscript at this stage. Please do not hesitate to contact us if you have any concerns or questions.

Yours sincerely,

Hyung Don Ryoo

Guest Editor

PLOS Genetics

Gregory P. Copenhaver

Editor-in-Chief

PLOS Genetics

Reviewer's Responses to Questions

**Comments to the Authors:**

Reviewer #1: In their article Wang and colleagues investigate the role of 2 ER resident E3 ubiquitin ligase SORDD1 and SORDD2 in retinal degeneration induced by Rh1G69D mutation, a model of adRP. By using a gain of function screen in animal expressing Rh1G69D, they have identified SORDD1/2 as suppressor of retinal degeneration. They further show that overexpression SORDD1/2 reduce ER stress and promote the degradation of Rh1G69D. Genome wide RNAi revealed that SORDD1 function requires VCP and the proteasome but not Hrd1 complex. Finally, crispr generated deletion of SORDD1/2 show no phenotype on its own but combined with hrd1 mutant exhibited enhanced degeneration and photoreceptor dysfunction in Rh1G69D mutant. This suggests that SORDD1/2 work in parallel with Hrd1 to degrade Rh1G69D.

The paper is interesting and the data are novel. However, some of the claims are not sufficiently supported by the data. In several cases, I was not fully convinced by the data provided and several data are not quantified.

Major comments

In figure 1, the authors state that Rh1G69D is overexpressed weakly by GMR promoter. I am surprised by this statement because GMR is a strong pan retinal driver expressed not only in photoreceptors but in all retinal cells. The authors should support their claim by showing expression level.

In figure 1b-k, the level of protection is not quantified here. Eye size and photoreceptor number should be quantified. The authors should also include a control with an overexpression control line such as UAS-GFP.

What is the contribution of retinal pigment cell versus photoreceptor in this phenotype.

Interestingly SORDD1 is expressed ubiquitously but its role seems restricted to photoreceptor cells. Could SORDD1 play a role in other cell types or other misfolded proteins?

At this stage, there is not enough evidence to claim that SORDD1 act in the ERAD complex. All is based on the localization at the ER with calreticulin. However the images provided does not allow to see if there is any cytoplasmic localization of SORDD1/2. These ubiquitin E3 ligases could simply be required for ubiquitination and proteasome function in the cytoplasm. What is the global protein ubiquitination profile on SORDD1/2 mutants?

Figure 2a should be quantified.

Figure 3b, c. Photoreceptor degeneration can sometimes undergo below an intact cornea so the authors should show photoreceptor count for these conditions.

Figure 4c. Deep pseudopupil can be difficult to distinguish in particular for quantification. Please show raw images or use an alternative for quantification such as cornea neutralization technique.

Minor comments

In first paragraph the authors use "glassy eye phenotype" I think we use glossy instead.

Reviewer #2: The submitted work by Xu et al. proclaimed the identification of novel E3 ligases, SORDD1/2, facilitating Rhodopsin 1 (Rh1) homeostasis in a Drosophila model and prevent mutant Rh1 induced retinal degeneration. Several rh1 mutations that fail to be correctly folded and transported to the apical membrane of photoreceptor cells could accumulate in the ER and trigger ER stress. The authors provide some evidence to argue that SORDD1/2 function in parallel with the HRD1 E3 ligase, whose action and other cofactor help mark the ubiquitin chains and provide a channel to allow the retrotranslocation of that misfolded rhodopsin to the cytoplasm. While this study's finding might be potentially significant, several issues need to be resolved before publication.

Major points:

1. The most puzzling question in the paper is that their finding contradicts to a prior report by Gricius et al. (2010) PLoS Genet. 6(8):e1001075. In that paper, authors used Rh1P37H to model a common ADRP mutation that mimicking a typical class II rod opsin mutation P23H, which is misfolded in the ER and processed by ERAD. However, they showed that by knocking down VCP/TER94, or by pharmacological treatment of either VCP or proteasome inhibitors, the retinal degeneration caused by Rh1P37H was suppressed. Although this study is using a different ADRP allele, the effect of VCP/TER94 is the opposite. Given that both alleles showed ER stress, the discrepancy shall be resolved experimentally. The authors should test Rh1P37H allele to demonstrate that SORDD1/2 are indeed genuine E3 ligases that function to process Rh1. Importantly, the authors shall bring this issue in the discussion. Furthermore, the only result that links TER94 function in SORDD1-mediated Rh1 degradation is a single RNAi allele analysis in Figure 3b, which is insufficient. TER94 has been shown to interact with several ER membrane components, the authors shall evaluate additional TER94 alleles and test if TER94 could physically interact with SORDD1.

2. Another concern is the data used to prove that SORDD1 is an ER membrane protein (Figure S1b). As this study claims, SORDD1 is an ER membrane component for the first time, and there is no prior report of this E3 ligase, robust analyses are required for scrutinizing. The image of the S2 cell shown in S1b has some caveats. First, the manuscript didn’t mention how eGFP was tagged to SORDD1. Second, the authors didn’t prove that the localization of SORDD1-GFP fusion protein remains the same as the endogenous SORDD1. Third, the GFP signals from the fusion protein and anti-Calnexin staining were exceedingly saturated. It is risky to use that overlapping to suggest SORRD1 is an ER membrane component. To convince readers that this E3 ligase indeed resides in the ER membrane, an antibody that could recognize the endogenous SORDD1 is preferable. Alternatively, if the antibody is not available or not work, it is necessary to provide either co-IP or proximity-based labeling results with a smaller tag to demonstrate that SORDD1 is indeed associated with a prominent ER protein.

3. The manuscript indicated that overexpression of E3 ligase CG8141 did not suppress Rh1G69D-induced eye phenotype. As both CG8141 and sordd1 are orthologous to RNF185 and RNF5, it will be more convincing to show that CG8141 has functionally deviated from sordd1/2 on the issue of Rh1 processing. The ERAD reporters used in Figure 2a should include GMR>Rh1G69D+ CG8141.

4. The manuscript underscores SORDD1/2 E3 ligase activity involves in Rh1 homeostasis. Interestingly, Figure S2a showed that SORDD1 expression still partially suppresses Rh1G69D-KR-induced rough eye phenotype. As the authors noted that Rh1G69D-KR is ubiquitin-resistant, several issues need to be clarified for this data. First, the authors didn’t mention what lysine residues of this transgenic fly have been changed. The detail of generating this ubiquitin-resistant allele should be included. Second, an IP experiment with anti-ubiquitin detection is essential to demonstrate that this KR allele is free from ubiquitination. Third, because the authors indicated that Rh1G69D-KR is ubiquitin-resistant, the authors should explain how does SORDD1 remains capable of the partial rescue?

5. The authors should experimentally clarify whether the modulation of sordd1 would affect endogenous hrd1 expression or vice versa, given that these two E3 ligases serve in a complementary manner.

6. There are several issues in those compound eye micrographs. First, I am troubled by the scale bar throughout the manuscript. If the scale bar is correct, the A-P dimension of a fruit fly eye is greater than 3 mm, which is impossible. Second, the micrographs of the same genotype often showed inconsistent morphology. For example, while both Figure 1d and Figure 2d showed the external eye morphology of GMR>Rh1G69D+hrd1, one can tell that in Figure 1d showed mild roughness, but the panel showed the same genotype in Figure 2d looked normal. Another example has to do with GMR>Rh1G69D+sorrd1 (Figure 1e) or sorrd2 (Figure 1f), which showed normal external eye morphology. However, when looking at the same genotype from the corresponded panels in Figure 2d, the rough eye phenotype was evident. Such irregularity of phenotype from the same fly strain raises great concerns, especially for interpreting the issue regarding hrd1 and sorrd1 function in parallel to processing Rh1, as shown in Figure 3c. The authors shall adopt methods with either a better resolution or with a quantifiable measurement to improve the quality.

7. The IP data in Figure 2e suggests the expression of SORRD1C165S could lower the ubiquitination level of Rh1G69D-Myc. However, it is apparent that anti-Myc IP result also showed much lower Rh1G69D-Myc protein levels. Because the experiment was performed in rpn12 knockdown background, the authors were awarded this problem, but chosen this data to argue the E3 ligase activity of SORRD1 is crucial for Rh1 processing does raise some concerns. The authors should provide data with a comparable amount of Rh1G69D-Myc in wild type SORRD1 and SORRD1C165S background to substantiate their findings.

8. One highlighted point in the manuscript is that SORDD1/2 and Hrd1 function in parallel and complementary to maintain Rh1 homeostasis. While the authors showed that loss of all three genes leads to age-dependent photoreceptor cell degeneration (Figure 4d), it is not clear whether this is due to the disturbance of Rh1 homeostasis. Furthermore, the main text or the figure legend didn't indicate whether these panels in Figure 4d are from wild type or Rh1G69D backgrounds. If this is from the wild type background, the authors should compare the endogenous Rh1 protein levels right before the onset of photoreceptor degeneration to address this issue.

9. Figure 1a is incorrect because the EP lines of SORDD1/2 are in the X chromosome. This kind of error shall avoid.

Minor points:

1. In the first Result section, the authors stated – "Rh1G69D was weakly expressed in undifferentiated photoreceptor neurons (GMR>Rh1G69D)"– is misleading. First of all, GMR-Gal4 is considered a strong driver, and there is no quantitative index or means to make such a statement. Second, GMR-Gal4 drives UAS transgene expression in all cells of differentiating ommatidia posterior to the morphogenetic furrow, not limiting to photoreceptor neurons.

2. The molecular weight of Rh1G69D in Figure S2b and Figure 2b seems quite a different judging from the marked MWs. A major band (blow 37 kD mark) could be found in the third and the fourth lanes of S2b. Is that an unspecific band or a processed Rh1?

3. Figure 2e and 2f appeared in the text without mention anything about rpn12, which only seemed later. The writing flow should be fixed.

4. Toward the end of the first paragraph in the third section of the Result, it should be “when knocked down alone” instead of “when expressed alone” because the corresponded experiments were using the RNAi approach.

5. The authors should provide a citation of experimental data to demonstrate that rpn12 knockdown would affect proteasome function.

6. The authors shall avoid using “cell death” in their text because none of the provided data includes any cell death assay. An example could be found to the end of section 3 in the Result.

7. The genotypes in Figure 3b and 3c are confusing. While the labeling of sordd1 was no difference in all panels, the genotypes stated in the Figure legend were either UAS-sordd1 or simply sordd1 (is this indicating the EP line?). The authors need to make sure the genotype is correctly described.

Reviewer #3: The manuscript by Jaiwei Xu, Haifang Zhao, and Tao Wang used an overexpression screen to identify two redundant ubiquitin ligases SORDD1/2 as additional factors that contribute to the degradation of misfolded opsins. By-and-large the experiments are well controlled and the data convincing. Identification of this novel HRD1/3 independent ER degradation pathway is intriguing and an important. I have just a few comments that should be straightforward to address.

1. For the ERG analysis (fig. 5) the flies were maintained under constant light.

It seems important to note that constant light treatment triggers elevated ER stress (PMID: 30015618). Is that elevated ER stress necessary to achieve the strong ERG phenotype reported? Was the same pre-treatment used for the deep pseudo pupil analysis?

2. The title is a bit misleading, as it is only the overexpression of SORDD1 that results in suppression of the retinal degeneration phenotype. Related to this; it may be old-fashioned, but the name SORDD1/2 seems a bit unfortunate. Classically, fly genes have been named for their loss-of-function rather than their overexpression phenotypes.

3. I do not understand why HRD1-independence of SORDD1 activity disproves that "that HRD1 may work as the retro-translocation channel for this kind of multi-span membrane substrate". author statement). It just seems to prove that it is not the only one especially given that the endogenous HRD1 and SORDD1 genes appear to be part of two redundant pathways for Rho[G69D] degradation (Figure 4)

4. There is a significant literature on the mammalian RNF5 / RNF185 homologs of SORDD1/2, for example in the context of cystic fibrosis or immunity. Given the functional conservation shown by the authors, known functions of these mammalian homologs should be discussed and compared to the current findings.

5. Regarding the Western blots shown in Figure 2e: Elevated ubiquitination of Rh1G69D–Myc in the SORDD1 Wt compared to SORDD1-C165S expressing flies is clear, but how do the authors explain that Rh1G69D–Myc is not accumulating to the same level given that proteasomal degradation is inhibited. This should be addressed.

Minor details:

6. The EM procedure mentions 4% paraformaldehyde and 2.5% pentanediol as fixative

Presumably the authors mean pentanedial more commonly referred to as glutaraldehyde.

7. lots of typos and grammar errors e.g.:

- Amalgam (Ama) one of gene identified in RNAi screen (Figure legend S2)

- Presumably statistical significance was assessed rather than "applied" (Statistics section)

- SORDD1 function dependently on VCP and the proteasome but independently on the RD1 complex (section 3)

- Degrading misfolded ER membrane proteins, as called ERAD-M,

- Mutations in hrd1 or hrd3 have no effect in eye morphology

**Have all data underlying the figures and results presented in the manuscript been provided?**

Reviewer #1: Yes

Reviewer #2: Yes

Reviewer #3: Yes

PLOS authors have the option to publish the peer review history of their article (what does this mean?). If published, this will include your full peer review and any attached files.

Reviewer #1: No

Reviewer #2: No

Reviewer #3: No

---

## [Decision Letter · Decision Letter 1]

28 Sep 2020

Dear Dr Wang,

Thank you very much for submitting your Research Article entitled 'Suppression of retinal degeneration by two novel ERAD ubiquitin E3 ligases SORDD1/2 in Drosophila.' to PLOS Genetics. Your manuscript was fully evaluated at the editorial level and by independent peer reviewers. The reviewers appreciated the attention to an important topic but identified some aspects of the manuscript that should be improved.

We therefore ask you to modify the manuscript according to the review recommendations before we can consider your manuscript for acceptance. Your revisions should address the specific points made by each reviewer.

[LINK]

Yours sincerely,

Hyung Don Ryoo

Guest Editor

PLOS Genetics

Gregory P. Copenhaver

Editor-in-Chief

PLOS Genetics

Dear Dr. Wang,

Thank you for submitting a revised version of your manuscript entitled 'Suppression of retinal degeneration by two novel ERAD ubiquitin E3 ligases SORDD1/2 in Drosophila' to PLOS Genetics. Your manuscript was evaluated by two of the three original referees. I also read your manuscript to evaluate your responses.

As you see, the reviewers found that your revised manuscript addresses most of the concerns that they had raised. You will also find that there are three relatively minor points that they ask you to address before publication. These are:

1. Reviewer 2 is asking why a panel in Figure 2a (that over expressing CG8141 has bright GFP signal that is not reflected in the quantified graph in Figure 2b. It appears to me that the specific Figure that you displayed shows episode-fluorescence and is not a representative one. Please choose a representative image to replace that panel.

2. Reviewer 2 also asks you to state in the main text that Rh1 G69D-KR has three lysine residues in the transmembrane region.

3. Reviewer 3 insists that you cannot conclude ER localization of SORDD1/2 based on over expression experiments. In my view, this could be readily addressed by deleting conclusions based on your own over expression experiments, and instead, cite other studies concluding that the mammalian SORDD1/2 homolog acts in the ER. 

Yours sincerely,

Hyung Don Ryoo

Reviewer's Responses to Questions

**Comments to the Authors:**

Reviewer #2: The revised manuscript by Xu et al. has made significant improvements. However, the following issues need to be resolved before acceptance for publication.

1. The authors have responded to my earlier concern regarding the functional deviation of SORDD1/2 and another E3 ligase CG8141. The revised manuscript has included the CG8141 overexpression allele in ER stress reporters assay. As seen in Figure 2a, the signals of XBP1-EGFP and ATF4-mCherry in the bottom row (Rh1G69D-Myc + CG8141) appeared significantly enhanced and did not follow Rh1G69D-Myc's staining pattern as compared to Rh1G69D-Myc alone (2nd row from top). Because CG8141 overexpression did not worsen eye degeneration, how do the authors explain this observation? While the authors showed quantitative data in Figure 2b, given the intensity difference of the represented image in Figure 2a, and the N value was 4 in Figure 2b, the data require careful review.

2. According to the authors’ response, Rh1G69D-KR has three lysine residues within the transmembrane domain that were not mutated to Arginine, which might explain why the expression of sordd1/2 could still partially rescue the Rh1G69D-KR-caused rough eye phenotype. This information, while no clear mechanistic insight, should be revealed in the main text.

Reviewer #3: The authors have appropriately addressed the majority of concerns raised by the previous round of reviews. The exception is their conclusion that SORD1/2 are ER proteins.

This is entirely based on the analysis of overexpressed SORD1/2 proteins in S2 cells. Accumulation of overexpressed proteins in the ER is not unusual. Therefore, the authors' conclusion that endogenous SORD1/2 are ER proteins is not convincingly supported by the data and the conclusion should be modified accordingly.

**Have all data underlying the figures and results presented in the manuscript been provided?**

Reviewer #2: Yes

Reviewer #3: Yes

PLOS authors have the option to publish the peer review history of their article (what does this mean?). If published, this will include your full peer review and any attached files.

Reviewer #2: No

Reviewer #3: No

---

## [Editor Report · Decision Letter 2]

5 Oct 2020

Dear Dr Wang,

We are pleased to inform you that your manuscript entitled "Suppression of retinal degeneration by two novel ERAD ubiquitin E3 ligases SORDD1/2 in Drosophila." has been editorially accepted for publication in PLOS Genetics. Congratulations!

Yours sincerely,

Hyung Don Ryoo

Guest Editor

PLOS Genetics

Gregory P. Copenhaver

Editor-in-Chief

PLOS Genetics

**Data Deposition**

http://datadryad.org/submit?journalID=pgenetics&manu=PGENETICS-D-20-00745R2

**Press Queries**

---

## [Editor Report · Acceptance letter]

22 Oct 2020

PGENETICS-D-20-00745R2 

Suppression of retinal degeneration by two novel ERAD ubiquitin E3 ligases SORDD1/2 in Drosophila. 

Dear Dr Wang, 

We are pleased to inform you that your manuscript entitled "Suppression of retinal degeneration by two novel ERAD ubiquitin E3 ligases SORDD1/2 in Drosophila." has been formally accepted for publication in PLOS Genetics! Your manuscript is now with our production department and you will be notified of the publication date in due course.

With kind regards,

Jason Norris

PLOS Genetics

On behalf of:
